# A Scalable Histological Method to Embed and Section Multiple Brains Simultaneously

**DOI:** 10.3390/cells13100860

**Published:** 2024-05-17

**Authors:** Divine C. Nwafor, Stanley A. Benkovic, Briana L. Clary, Allison L. Brichacek, H. Wayne Lambert, Matthew J. Zdilla, Candice M. Brown

**Affiliations:** 1Department of Neurosurgery, University of Virginia, Charlottesville, VA 22903, USA; jgx9as@uvahealth.org; 2Department of Neuroscience, School of Medicine, West Virginia University, Morgantown, WV 26506, USAbc00022@mix.wvu.edu (B.L.C.); 3Rockefeller Neuroscience Institute, West Virginia University, Morgantown, WV 26506, USA; 4Department of Microbiology, Immunology, and Cell Biology, School of Medicine, West Virginia University, Morgantown, WV 26506, USA; allison.brichacek@nih.gov; 5Department of Pathology, Anatomy and Laboratory Medicine, School of Medicine, West Virginia University, Morgantown, WV 26506, USA; hwlambert@hsc.wvu.edu (H.W.L.); matthew.zdilla@hsc.wvu.edu (M.J.Z.)

**Keywords:** brain, gelatin, histology, microscopy, sectioning

## Abstract

The preparation and processing of rodent brains for evaluation by immunohistochemistry is time-consuming. A large number of mouse brains are routinely used in experiments in neuroscience laboratories to evaluate several models of human diseases. Thus, methods are needed to reduce the time associated with processing brains for histology. A scalable method was developed to embed, section, and stain multiple mouse brains using supplies found in any common histology laboratory. Section collection schemes can be scaled to provide identical bregma locations between adjacent sections for immunohistochemistry, facilitating comprehensive, high-quality immunohistochemistry. As a result, sectioning and staining times are considerably reduced as sections from multiple blocks are stained simultaneously. This method improves on previous procedures and allows multiple embedding and subsequent immunostaining of brains easily with a dramatically reduced time requirement. Furthermore, we expand this method for use in numerous mouse tissues, rat brain tissue, and post-mortem human brain and arterial tissues. In summary, this procedure allows the processing of many rodent or human tissues from perfusion through microscopy in 10 days or less.

## 1. Introduction

Immunohistochemical analysis of experimental brains can be an expensive and time-consuming procedure. Processing of brains from perfusion through staining may take several weeks. Additionally, antibodies and associated staining equipment are expensive and can cost several thousand dollars to perform a basic experiment. These reasons make it difficult for small neuroscience laboratories to consider histology as a method to assess neuronal damage, gliosis, and immune system activation in routine experiments. To address this issue, scalable and efficient histological methods that process a large number of brains in a timely manner using standard laboratory equipment are desperately needed.

The most time-consuming procedures in histological analysis are sectioning and staining, and a method to streamline these processes would save considerable time. Several laboratories have developed systems to embed multiple brains into a matrix, and section and stain representative subsets simultaneously. A recent protocol described a megabrain technique utilizing an Optimal Cutting Tissue compound (OCT) to expedite tissue processing and staining. However, the megabrain matrix was extremely temperature-sensitive and was mostly suitable for slide-staining sections [1]. Gelatin was initially used as a stabilizing matrix [2] and was later used as an embedding matrix, although the details were not fully described [3]. A gelatin–albumin mixture has also been developed and used successfully to embed, section, and stain rodent and human brains [4]. Another method to align and embed whole mouse brains or hemispheres in a gelatin-based “receiving matrix” has been developed and used to embed, section, and stain multiple mouse brains in different anatomical orientations [5]. While these previously published methods have produced satisfactory results in many laboratories, they are also time-consuming and require an intermediate level of technical expertise.

We describe a method for embedding, sectioning, and staining multiple mouse brains with an inexpensive procedure using common laboratory supplies such as microscope glass slides. We also offer an alternative embedding form if a laboratory has access to a 3D printer. The procedures described here facilitate the processing of experimental brains from perfusion through microscopy in approximately 10 days or less. Furthermore, we demonstrate that our method is scalable to various tissue types in rodents and humans.

## 2. Materials and Methods

### 2.1. Animal Breeding and Housing

All procedures were approved by the West Virginia University Animal Care and Use Committee. Mice were bred in the vivarium of the West Virginia University Health Sciences Center. Animals were maintained on a 12:12 light/dark cycle with access to food and water ad libitum. Wild-type (WT; C57BL6/J; Bar Harbor, ME, USA, Catalog # 000664) mice were used in this study. Rat brain tissue was obtained from the laboratory of Dr. Mark Olfert at West Virginia University.

### 2.2. Human Brains

The West Virginia University Human Gift Registry provided post-mortem de-identified human brain tissues used in this study. The West Virginia State Anatomical Board approved this research, and due to the use of cadaveric material from the West Virginia Human Gift Registry, this study was exempt from the Institutional Review Board.

### 2.3. Tissue Collection and Processing

Deeply anesthetize the mouse with isoflurane and perfuse intracardially with a perfusion pump (Masterflex 7524-10, Cole-Parmer, Vernon Hills, IL, USA) set to 5.0 mL/min. Rinse blood from the vasculature with 20 mL 0.9% saline and fix the brain with 50 mL 4% paraformaldehyde (PFA) (Fisher Scientific, Pittsburgh, PA, USA). Remove the mouse brain from the skull and post-fix in 4% PFA overnight at 4 °C. Human brain and vessels, rat brain, mouse liver, spinal cord, and intestines were also post-fixed in 4% PFA. The next day, rinse the tissues in 0.01 M phosphate-buffered saline (PBS) (Fisher Scientific, Pittsburgh, PA, USA) and incubate sequentially in 15% and 30% sucrose in PBS for 24 h each (or until the tissues in sucrose solution sink to the bottom of the storage vessel). NOTE: Intestinal tissues may take longer to sink.

### 2.4. Gelatin Solution

Make a solution of 15% gelatin (Type A, strength ~300 g bloom, Sigma-Aldrich, St. Louis, MO, USA) in PBS by adding 15 g gelatin powder to a bottle, pouring 100 mL 0.01 M PBS over the powder, and incubating at 42 °C in a water bath until the powder dissolves. Maintain the solution at 42 °C until the embedding step is complete.

### 2.5. Embedding Form Construction

#### 2.5.1. Microscope Slides

Construct an embedding form using five standard microscope slides (Fisher Scientific, Pittsburgh, PA, USA). Form the slides into an open-topped rectangle and hold them in position with laboratory tape (Figure 1A). Cool the mold on a cold plate at 4 °C (TCP-2, ThermoElectrics Inc, Texas city, TX, USA.) and seal the intersections of each slide with gelatin solution (Figure 1B). The cold plate allows the gelatin to harden.

#### 2.5.2. Three-Dimensional Printer

Construct a rectangular embedding form using a 3D printer that fits into a recessed base (Figure 1A). Seal the intersections of the form and the base using the gelatin solution.

### 2.6. Embedding and Post-Processing

On the day of embedding, transfer the brains into a 15 mL gelatin solution and incubate at 42 °C for 1 h. Pour a 2mm gelatin base into form and allow it to harden on the cold plate at 4 °C (Figure 1C). Remove the brains quickly from the warm gelatin solution and place them in the embedding mold on the cold plate (at 4 °C) in the orientation selected for sectioning. The brains were added with forceps or a spatula (Figure 1D) and aligned in the pattern shown by the block map (see example in Figure 2C). Take care to align the brains in the same direction unless one was intentionally turned to provide a block map (Figure 1D and Figure 2C). For the horizontal plane, embed six brains in order to fit on a standard frosted-end slide. If unfrosted slides are available, eight brains can be embedded. After all brains are in place, allow 5 min to pass to promote the binding of the gelatin-covered brains to the base. Carefully fill the mold with warm gelatin to approximately 3 mm over the top of the brains. Allow the gelatin to harden for 5 min and transfer the block to the refrigerator for 1 h at 4 °C (Figure 1E and Figure 2A)

Disassemble the mold. Trim any excess gelatin and return to the stock bottle for reuse (Figure 1F and Figure 2B). Process the gelatin brain block sequentially in 4% PFA for 24 h, 15% sucrose for 24–48 h, and 30% sucrose for 48 h. NOTE: the 15% sucrose step may be skipped if emergent staining is required.

### 2.7. Sectioning

Section on a sliding microtome (HM 450, ThermoFisher Scientific, Waltham, MA, USA) equipped with a 3 × 3 freezing stage (BFS-40MPA, Physitemp, Clifton, NJ, USA). On the day of sectioning, place the block in a −80 °C freezer for 1 h. Freeze-adhere the frozen block to the stage using OCT (TISSUE TEK, Sakura Finetek, Torrance, CA, USA) (Figure 3A). Obtain sections at different thicknesses, e.g., 25, 35 or 50 µm intervals in the sagittal or horizontal plane and collect into a series of specimen cups (Glad minirounds, 4oz, Kroger, Cincinnati, OH, USA). NOTE: We recommend 30–35 µm tissue sections. Thinner gelatin tissue sections are difficult to mount on slides.

### 2.8. Collection Strategy

Sequentially place sections into specimen cups filled with 0.01 M PBS (Figure 3B). Note: three 6-well plates can be used instead of the specimen cups shown in this protocol. Collect sections into PBS for short-term storage (<2 months) or PBS plus 0.065% sodium azide for long-term storage at 4 °C (>1 year). Sections can also be transferred into a cryoprotectant solution (30% ethylene glycol, 30% sucrose in 0.1M PBS) for long-term storage at 4 °C or −20 °C.

### 2.9. Histochemical and Immunohistochemical Staining

#### 2.9.1. Brightfield Immunohistochemistry

Stain free-floating sections using a modified ABC procedure (Vector Laboratories, Burlingame, CA, USA). Treat sections with 10% methanol, 10% hydrogen peroxide in 0.01 M Dulbecco’s modified PBS (DPBS; 136 mM NaCl, 8 mM Na_2_HPO_4_, 2.6 mM KCl, 1.5 mM KH_2_PO_4_) for 15 min to quench endogenous peroxidase. Following three rinses in DPBS for five min each, incubate sections in a permeabilizing solution (4% normal horse serum, 0.2% Triton X-100 in DPBS) for 30 min at room temperature (RT). Transfer sections directly into primary antibody solution in DPBS + 4% normal horse serum and incubate overnight at RT (see antibody dilutions in Table 1). The following day, rinse sections three times in DPBS for 5 min each and transfer into secondary antibody solution in DPBS + 4% normal horse serum for two hours at room temperature. Following three rinses in DPBS for 5 min each, incubate sections in Avidin D-HRP (1:1000 in DPBS, Vector Laboratories, Burlingame, CA, USA) for 1 h at room temperature, rinse three times in DPBS for 5 min each, and incubate with chromogen solution (3-3′ diaminobenzidine, 50 mg in 100 mL DPBS + 50 µL 30% hydrogen peroxide, Electron Microscopy Sciences, Hatfield, PA, USA) for 5 min. Rinse sections three times in DPBS for 5 min each, mount onto microscope slides (Colorfrost +, Fisher Scientific, Pittsburgh, PA, USA), allow to air-dry at RT for two days, dehydrate through a standard dehydration series, and coverslip with Permount (Fisher Scientific, Pittsburgh, PA, USA). NOTE: a slide warmer can be used to expedite the drying process to 1–2 h instead of air-drying for two days at RT.

#### 2.9.2. Fluorescent Immunohistochemistry

Follow a similar procedure for the first day of brightfield detection. On the second day, rinse sections three times in DPBS for 5 min each and transfer sections into fluorescent-tagged secondary antibody solution in DPBS for 2 h at RT in the dark. Rinse sections three times in DPBS for 5 min each, and incubate in Hoechst 33342 (1:10,000 in DPBS, H3570, ThermoFisher Scientific, Waltham, MA, USA) for 1–2 min. Rinse sections three times in DPBS for five min each, mount onto microscope slides, and coverslip with Vetashield antifade mounting medium (H-1000-10, Vector Laboratories, Burlingame, CA, USA). 

#### 2.9.3. Perl’s Prussian Blue (Microhemorrhages)

Evaluate the detection of iron with a slightly modified procedure [6]. Mount sections onto microscope slides (Colorfrost +, Fisher Scientific, Pittsburgh, PA, USA) and allow to air-dry at RT for two days. NOTE: a slide warmer can used to expedite the drying process to 1–2 h instead of air-drying for two days at RT. Dip slides in distilled water and then transfer slides to fresh Perl’s solution (1% potassium ferrocyanide, pH = 1, 5% polyvinylpyrrolidone (PVP) for 60 min with shaking. Rinse slides three times in PBS for 5 min each and incubate in methanol containing sodium azide (0.01 M NaN_3_ + 0.3% H_2_O_2_) for 60 min with shaking. Rinse slides three times in PBS for 5 min each and counterstain with eosin for 1 min. Rinse slides three times in PBS for 5 min each, air-dry at RT for two days, dehydrate through a standard series, and coverslip with Permount. NOTE: A slide warmer can be used to expedite the drying process to 1–2 h instead of air-drying for two days at RT.

#### 2.9.4. Nissl Staining

Evaluate general morphology by standard Nissl staining. Mount sections onto microscope slides and allow to air-dry at RT for two days. NOTE: a slide warmer can be used to expedite the drying process to 1–2 h instead of air-drying for two days at RT. Dip slides in distilled water and transfer to staining solution (0.2% cresyl violet + five drops glacial acetic acid) for 5 min. Rinse slides in distilled water and incubate in 70% ethanol for 1 min. Differentiate the stain in 95% ethanol, and dip the slides in 100% ethanol for 30 s. Transfer the slides to xylene (3× for 1 min each) and coverslip with Permount.

#### 2.9.5. Alkaline Phosphatase (AP) Enzyme Stain 

Evaluate AP activity by incubating sections with a BCIP/NBT detection kit (SK-5400, Vector Laboratories, Burlingame, CA, USA). Rinse and incubate free-floating sections in BCIP/NBT solution for 6–8 h. Rinse sections three times in DPBS for 5 min each, mount onto microscope slides, air-dry at RT for two days, dehydrate through a standard dehydration series, and coverslip with Permount. NOTE: a slide warmer can be used to expedite the drying process to 1–2 h instead of air-drying for two days at RT.

#### 2.9.6. Hematoxylin and Eosin (H/E) Stain

Evaluate general morphology by standard H/E staining. Mount sections onto microscope slides and allow to air-dry at RT for two days. NOTE: a slide warmer can be used to expedite the drying process to 1–2 h instead of air-drying for two days at RT. Stain in hematoxylin for 1 min, then wash with 4–5 changes of tap water. Wash in 1× PBS for 1 min followed by 3 changes in distilled water. Counterstain in eosin for 1 min. Dehydrate through a standard dehydration series, and coverslip with Permount.

### 2.10. Microscopy and Photography

Tissue sections were viewed on a Leica DM6B microscope (Leica Camera, Allendale, NJ, USA) and images were captured using Leica LASX software (Leica Microsystems, Buffalo Grove, IL, USA). Processing and assembly of images into the final figures was performed with Photoshop CC 2017 (2017.0.1, Adobe Inc., San Jose, CA, USA).

## 3. Results

### 3.1. Evaluation of Embedding Forms

Embedding forms made from microscope slides were used successfully to embed multiple brains in several different schemes (Figure 1A). Experiments with up to 24 mouse brains were split across four forms. The slides were assembled in position and secured by laboratory tape. The form was placed on a cold plate, the intersections of the glass slides were sealed with gelatin to prevent the leaking of warm gelatin solution in later steps (Figure 1B). Finally, a 2 mm base was poured and allowed to harden (Figure 1C). The brains were added with forceps or a spatula (Figure 1D) and aligned in the pattern shown by the block map (see example in Figure 2C). The gelatin was allowed to solidify for 5 min to promote attachment of the brain to the base, and the form was filled with warm gelatin solution to a level of 3 mm above the top of the brains (Figure 1E). The addition of the gelatin solution was facilitated by the clear glass of the slide, enabling the visualization of the level of gelatin. The block was allowed to solidify for 5 min and placed into the refrigerator (at 4 °C) to harden. Once solidified, the form was disassembled, the block was trimmed with a razor blade (Figure 1F), and the extra gelatin was returned to the bottle for reuse. The three microscope slides forming the long axis of the base were left taped together and all slides were washed to remove gelatin debris. The bases were used to embed five or six blocks before the tape needed replacement. 

Alternatively, a 3D printer may be utilized to create an embedding form (Figure 1A, left; Figure 2A, right). The chamber was inserted into a recess in the base. Next, the assembled form was placed on the cold plate, the intersections were sealed with gelatin solution, and a 2 mm gelatin base was poured and allowed to harden. The brains were added (according to the block map) and allowed to attach to the base, and the form was filled with warm gelatin solution to a level of 3 mm above the top of the brains. After hardening, a razor blade was run across the sides of the form to remove the gelatin block. The block was trimmed, and the excess gelatin was returned to the bottle for reuse. The form was washed to remove gelatin debris. 

### 3.2. Section Collection Strategy

The number of cups used to collect sections at the microtome may vary and depend on the number of stains being evaluated. We used six specimen cups in this protocol for tissue collection (Figure 3A,B). This provided an adequate number of sections to evaluate the immunohistochemistry throughout the brain and allowed an appropriate number of adjacent immunostains. Two collection cups are recommended for practice brains. If a greater number of stains were desired post-collection, we recommend that three 6-well plate dishes be used for tissue collection rather than the specimen cups shown here. 

### 3.3. Example Slide-Mounted Sections and Immuno- or Histochemical Stains

Blocking techniques and immunostains should be performed depending on the type of experimental treatment and desired outcome being investigated. Whole mouse brains or hemispheres were sectioned in the horizontal or sagittal plane (Figure 4). In experiments with a large number of experimental animals, all positions in the block are filled requiring one brain to be turned off-axis to provide an identity for a block map (Figure 2C). We demonstrated that our embedding model is suitable for both brightfield (Iba-1, Figure 5A; NeuN, Figure 5D) and fluorescence (GFAP, Figure 5B; NeuN, Figure 5C) immunohistochemistry. Furthermore, histochemical stains were performed to evaluate general morphology (Nissl, Figure 5E) or to evaluate iron content (Perl’s Prussian blue, Figure 5F). Enzymatic activity stains (alkaline phosphatase, Figure 5G) were also performed and were used in combination with brightfield immunohistochemistry for a CD31-vascular marker (Figure 5H). 

### 3.4. Scalability of the Gelatin Embedding Matrix

The gelatin embedding method can be used to examine brain tissue histology in other species besides mice. Figure 6A,B demonstrate brightfield GFAP staining of astrocytes in the cortex of rat and human brain tissue. Furthermore, the gelatin embedding method can be utilized in other tissues such as the intestine (CD31-vascular marker, Figure 6C), spinal cord (alkaline phosphatase, Figure 6D), liver (Nissl, Figure 6E), and human arterial vessel (Hematoxylin and eosin, Figure 6F). 

## 4. Discussion

In this protocol, we describe a scalable method utilized by our laboratory to embed tissues into a gelatin matrix. Most importantly, this method improves upon existing methods due to cost, ease of implementation, and time savings. The embedding forms used in these experiments were inexpensive and were mostly derived from common supplies found in any histology laboratory. The form made from microscope slides was easy to assemble, use, and disassemble. The 3D-printed form separated easily from the base and a razor blade was run over the internal sides of the form. Both forms were used successfully in a number of experiments that involved embedding different tissue types.

The gelatin base provided a common plane for embedding and also provided elevation from the freezing stage of the microtome. It was important to trim the brainstems to the same level; otherwise, the posterior part of the brain would sit higher than the anterior part when embedded in the horizontal plane. When embedding in the coronal plane, the cerebellum should be removed (we evaluate the cerebellum in the horizontal or sagittal planes if needed) to allow the brains to stand up more easily in the gelatin. Alignment of all brains in the same position (i.e., bregma point) is critical for making comparisons among tissue sections. This is easily achieved since all tissues are provided with a common gelatin base prior to embedding.

Typically, six brains are embedded per block in our laboratory. The identity of each experimental brain was recorded in the block map. Maps for experiments with more than six brains were produced with differing appearances by leaving a space unoccupied or by turning a brain in a different direction. This allowed group staining of sections from multiple blocks so that comparisons across blocks were possible. Differences in staining intensity due to slight variations in incubation times in staining solutions were kept to a minimum. This also aids microscopy since illumination levels and exposure times to capture images are more constant. If the mouse brains are in the coronal plane, up to 8–10 brains may be embedded per gelatin matrix. Alternatively, if the mouse brains were hemibrains in the coronal plane, up to 12–15 hemibrains can be embedded per gelatin matrix. 

Gelatin matrix fixation and cryoprotection are critical steps in this protocol to minimize bacterial growth, prevent ice crystal formation, and promote the preservation of gelatin integrity. The brain blocks were frozen by placing the block on a cap from a scintillation vial and placing the gelatin matrix in the −80 °C freezer for 1 h. A dry ice–isopentane mixture can also be used for rapid freezing for 20 s; however, the temperature must be between −45 and −50 °C to avoid cracking the gelatin matrix. We found that the best temperature range to section the gelatin matrix on the sliding microtome was between −20 and −25 °C. The number of collection cups can be scaled to accommodate different staining strategies. Generally, a six- or eight-cup collection scheme is used, but this can be expanded to any number of cups desired. We have used two (for practice brains), six, or eight cups (for experimental brains) most commonly. In an eight-cup collection strategy of a brain embedded in the horizontal plane sectioned at 50 µm, the first section went into cup one, and every section sequentially thereafter (the ninth section went into cup 1). This resulted in eight adjacent dorsal-ventral series of sections that were 400 µm apart (50 µm × 8). Staining was facilitated by using one cup of sections for each stain, resulting in eight adjacent series of immunohistochemical stains containing approximately 15 sections each. It is important to note that thicker sections, greater or equal to 30 µm, are easier to mount on microscope slides compared to thinner sections, less than 30 µm.

This procedure builds upon work from other laboratories but requires less effort and technical expertise to construct the embedding forms [4,5]. A recent method was described that used a primary brain array cast in the shape of eight murine whole brains or ten hemispheres [5]. A secondary “receiving matrix” was constructed from 15% gelatin and the experimental brains were secured in the form with gelatin solution. Incubation of the brains in gelatin solution prior to embedding provided better adhesion of the brains to the matrix and solved issues with brains falling out of the gelatin post-sectioning. This method allowed the brains to be aligned along a predetermined axis and produced high-quality sections and immunostains. Subsequent processing of the block and staining of the sections was comparable to our procedure and only differed in the method of embedding mold used, freezing, and the type of microtome used. In experiments containing multiple blocks, we turned one brain off-axis rather than notch a corner of the gelatin as this enabled more blocks to be stained simultaneously. Also, we demonstrated in this protocol that our method is scalable to various tissues such as mouse liver, mouse intestine, mouse spinal cord, rat brain, human brain, and human vessels. Moreover, sectioning time was reduced to 1/6 (or 1/8 if using unfrosted slides), e.g., 24 brains can be blocked and sectioned in 4 h versus 24 h (i.e., 1 h per brain on a sliding microtome). Staining time was also reduced greatly since sections from multiple blocks could be stained in one large batch. Overall, with the inclusion of the time required for immunohistochemistry or histology, this method can be completed in 10 days. It is also important to note that our sections are not sensitive to temperature changes, which is a major limitation of the OCT megabrain protocol. Additionally, experimental samples may be lost within minutes if OCT is stored at RT or the temperature is not maintained within a range of −45 °C to 65 °C [1].

In conclusion, we describe a convenient and accessible method for embedding, sectioning, and staining multiple brains using supplies found in any standard scientific laboratory. A major limitation of this method is the inability of the gelatin to maintain its integrity when thinner tissue sections (<26 µm) are needed. This limitation is secondary to the lack of stronger bonds between the tissue and the gelatin when sections are thin. If thinner sections are needed, we recommend mounting the sections on slides immediately after sectioning the gelatin block. We have found that thinner sections mounted within 6 weeks are less susceptible to this issue. Another limitation of this method is that as the size of the gelatin block increases, there is a slightly increased likelihood of generating freeze artifacts in tissues. Nevertheless, this technique allows for the rapid evaluation of experiments that utilize a large number of experimental animals to produce a series of adjacent stains and/or immunostains for experimental comparisons and pathological evaluation.

## Figures and Tables

**Figure 1 cells-13-00860-f001:**
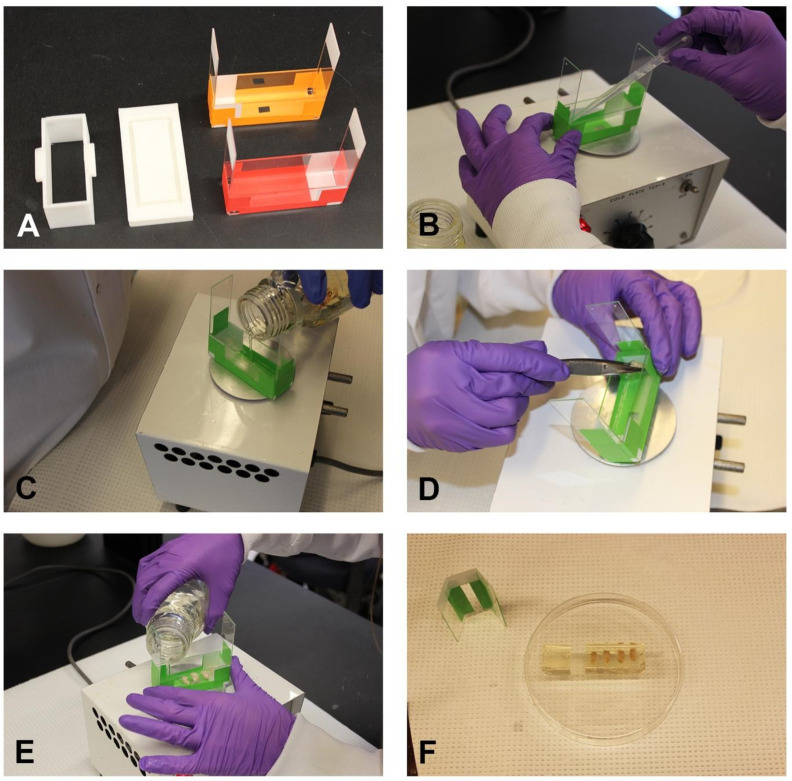
Embedding mouse brains using forms constructed from microscope slides. (**A**) Rectangular forms were constructed from five microscope slides or optionally on a 3D printer. (**B**) The form was placed on a cold plate and the intersections of the slides were sealed with gelatin solution. (**C**) A 2 mm gelatin base was poured and allowed to harden. (**D**) Brains were transferred from the gelatin solution at 42 °C to the form and allowed to set for 5 min. (**E**) Gelatin was carefully poured over the brains to a level of 2–3 mm and allowed to set for 5 min. (**F**) The block was placed in the cold room for 1 h, removed from the form, and trimmed to remove excess gelatin.

**Figure 2 cells-13-00860-f002:**
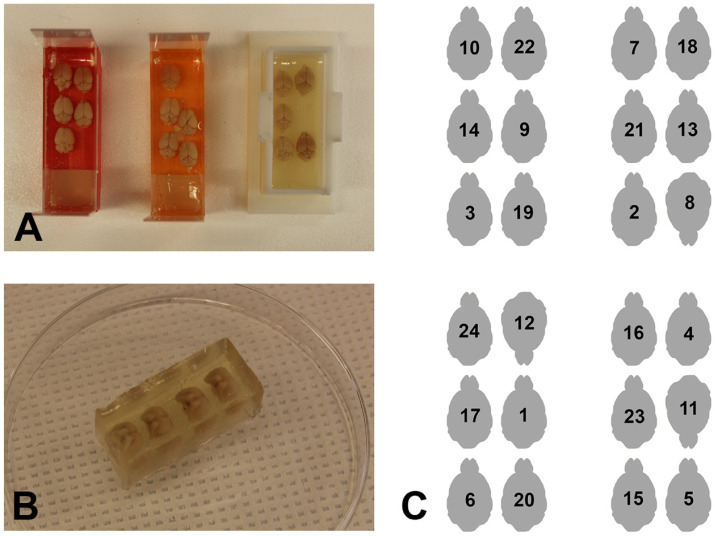
Block maps provide the identity of brains in experiments containing multiple blocks. (**A**,**B**) Frosted-end microscope slides and the 3D-printed form provide spaces for up to six mouse brains or four rat brains. In experiments requiring multiple blocks, a space may be left unoccupied, or one brain may be turned off-axis to provide identity for that block. (**C**) The block map for a 24-brain experiment is provided. Sections from all four blocks were stained simultaneously in the same dish since the sections were distinguishable from the map.

**Figure 3 cells-13-00860-f003:**
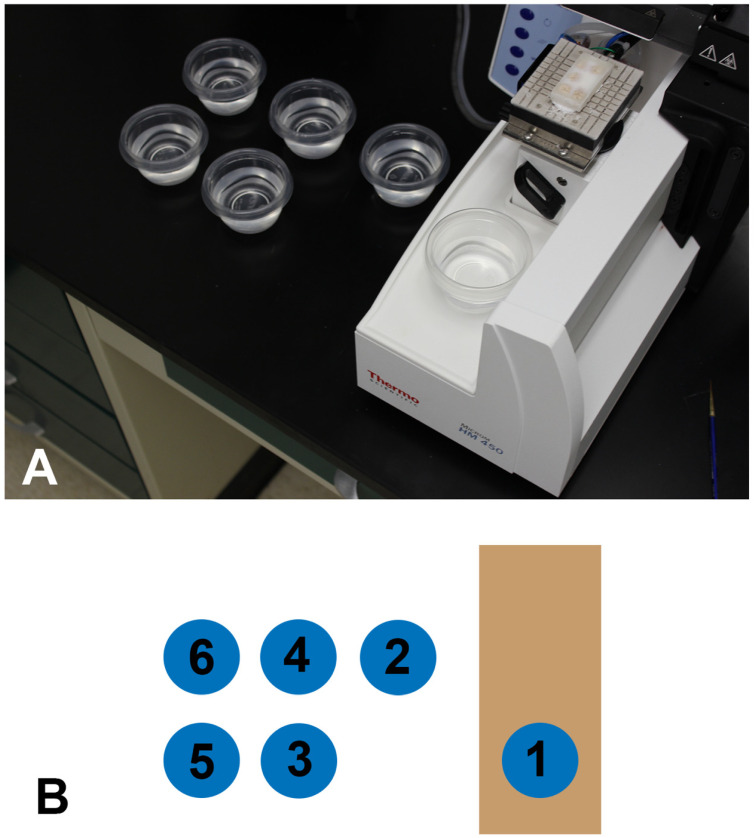
A scalable section collection scheme facilitates adjacent sets of stains. (**A**) Sections cut on the microtome were collected into a series of cups that considered the number and kind of stains required to be analyzed in the experiment. (**B**) The first section cut went into cup 1, the second section into cup 2, etc. The seventh section went into cup 1, and this pattern was repeated until the block was sectioned completely. For sections cut at 50 µ, this resulted in each cup containing a series of sections that were equally spaced at 300 µ (50 µ × 6). This collection scheme allowed the production of six sets of adjacent stains or immunostains.

**Figure 4 cells-13-00860-f004:**
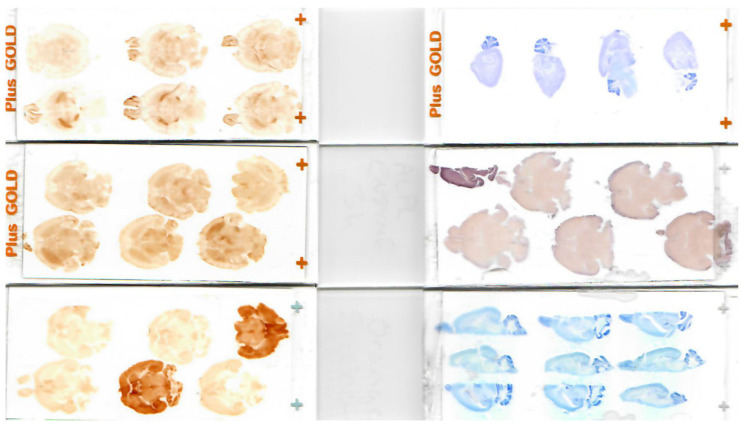
Example blocks and embedding schemes. A variety of whole brains or hemispheres embedded in various planes are shown. We generally section whole brains in the horizontal plane but have investigated hemispheres or other planes of section depending on the requirements of the experiment. These sections have been stained successfully with histochemical, immunohistochemical, or enzyme stains.

**Figure 5 cells-13-00860-f005:**
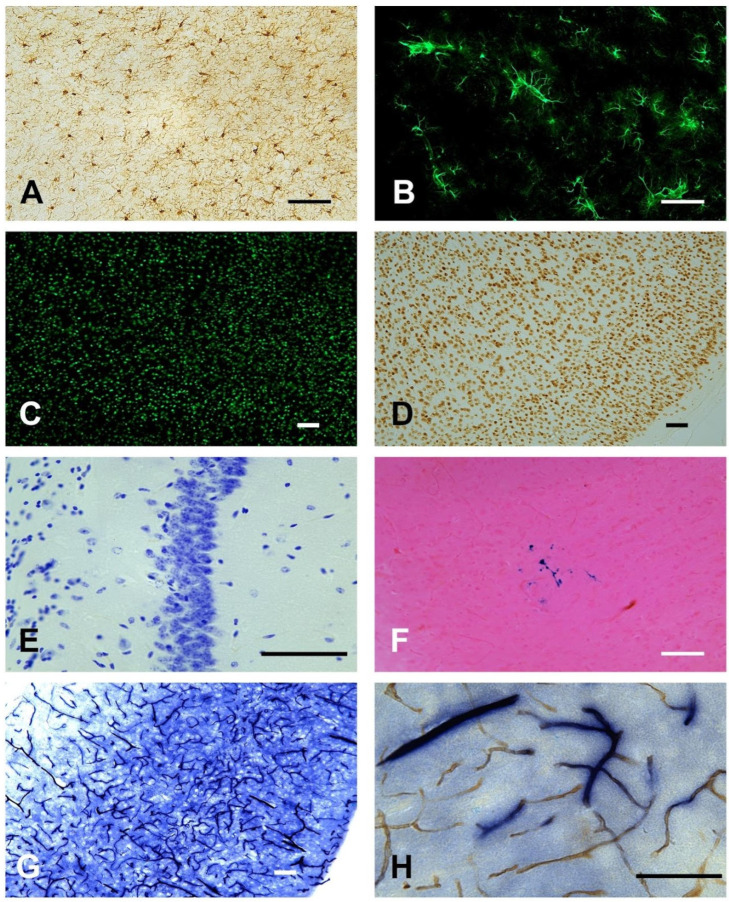
Examples of stains in mouse brain tissue produced using this method. A wide variety of stains, immunostains, and enzyme stains have been evaluated both by brightfield and fluorescent detection. (**A**) Microglial reactivity was evaluated by Iba-1 immunostaining and brightfield detection (brown). (**B**) GFAP immunolabeling for astrocytes (green) was evaluated by immunohistochemistry with fluorescent detection. (**C**,**D**) Neuronal cell body (NeuN) was evaluated by fluorescent (green) and brightfield (brown) detection. (**E**,**F**) Nissl (blue) and Perl’s Prussian Blue (blue) standard histochemical stains were used to evaluate general cell morphology and brain microhemorrhages, respectively. (**G**,**H**) Enzyme stains were used independently to evaluate alkaline phosphatase activity in blood vessels (blue) or in combination with immunohistochemistry for the endothelial antigen, CD31 (brown). Scale bar = 75 µm.

**Figure 6 cells-13-00860-f006:**
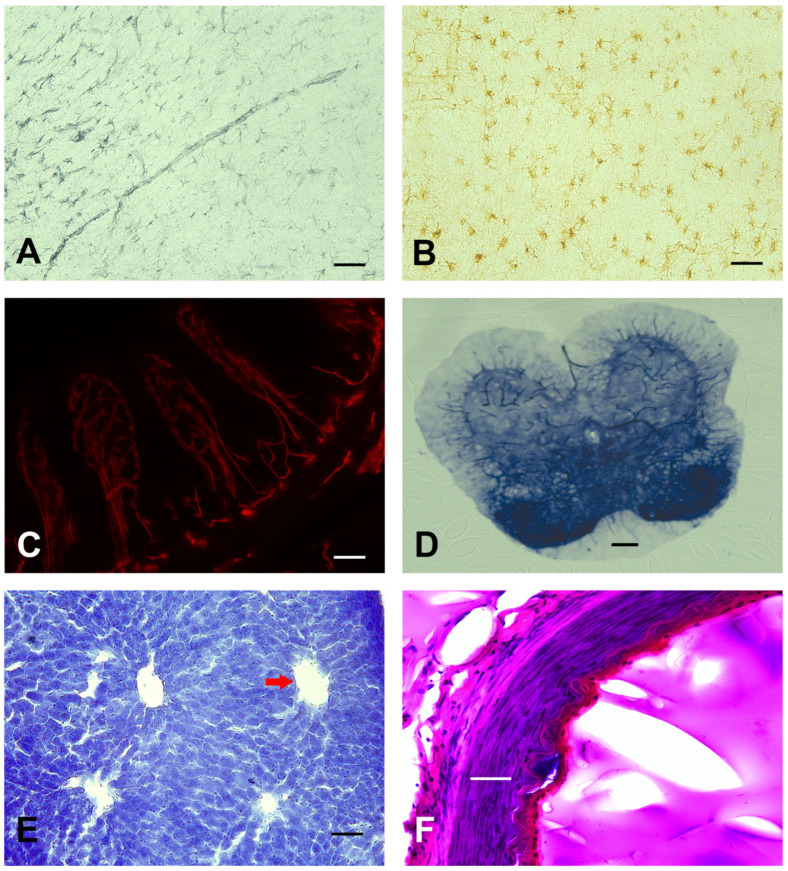
Scalability of the gelatin embedding matrix to various tissue types in different mammalian species. (**A**,**B**) GFAP (astrocytes) brightfield immunohistochemistry was evaluated in rat (gray) and human (brown) brain cortical tissue. (**C**) Murine intestinal tissue sections demonstrated blood vessel (CD31; red) fluorescent detection. (**D**) Enzyme stains were used to evaluate alkaline phosphatase activity in murine spinal cord sections. (**E**,**F**) Standard histological Nissl and H/E were used to evaluate general morphology in the murine liver (red arrow = central vein) and human arterial vessel, respectively. Scale bar = 50 µm (**A**–**F**).

**Table 1 cells-13-00860-t001:** Dilutions for antibodies used in immunohistochemical studies.

Antibody	Vendor	Primary	Secondary	Fluor-Secondary
GFAP	Dako	1:10,000	1:10,000	1:1000
Iba-1	Wako	1:2000	1:1000	
NeuN	CST	1:400	1:500	1:500
CD31	R&D Systems	1:500	1:500	1:500

## Data Availability

The original contributions presented in the study are included in the article, further inquiries can be directed to the corresponding author.

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
