# Peer review of "A Scalable Histological Method to Embed and Section Multiple Brains Simultaneously"

_cells, 2024, doi:10.3390/cells13100860_

Round 1

Reviewer 1 Report

Comments and Suggestions for Authors

The manuscript introduces a novel histological technique designed to embed and section multiple brains simultaneously, aiming to streamline processes in neuroscience research. This method significantly reduces the time required to prepare rodent and human brain tissues for immunohistochemical analysis, which is crucial for studying disease models. However, there are some concerns about several issues as noted below:

1. The paper mentions that this method allows for the processing of a large number of samples in a shorter period, but there is a lack of detailed operational instructions and data support regarding the comparability and uniformity between samples. How can one ensure consistent quality of sample processing under high-throughput conditions?

2. The method involves using gelatin as an embedding medium, which facilitates the simultaneous processing of multiple tissues. However, concerns have been raised regarding the integrity of gelatin sections containing thinner or irregularly shaped tissues. How does this affect the quality of histological sections, especially when detailed cellular structures need to be observed and analyzed?

3. The authors should elucidate the distinctions between gelatin and other embedding mediums in the Discussion section. Besides, it is crucial to highlight the advantages of using gelatin, providing a clear justification for its selection over alternative materials.

4. There are several formatting errors present in this manuscript. For instance, the symbol "" is incorrectly written as "C" (lines 84, 93, 94, 117, 125, etc.). Additionally, certain terms are inconsistently abbreviated, for example, "paraformaldehyde" is not abbreviated after its initial mention in line 84, while "RT" is abbreviated for the first time only in line 156, despite multiple prior occurrences. Please correct similar errors throughout the paper.

Author Response

  1. The paper mentions that this method allows for the processing of a large number of samples in a shorter period, but there is a lack of detailed operational instructions and data support regarding the comparability and uniformity between samples. How can one ensure consistent quality of sample processing under high-throughput conditions?

Rebuttal: We have utilized this method and published our findings in impactful journals such as BBI, Scientific Report, and BBR (PMID: 31778743, 31827139, 33137399). Other neuroscience groups have also utilized our method for publication in impactful journals and performed their image analysis independent of our group (PMID: 33992581, 33779614). We are confident that this method is reliable. The positioning of the brain at the same level with a gelatin base provides additional uniformity compared to conventional IHC, where each brain is sectioned one at a time, and uniformity is highly dependent on the user selecting the appropriate bregma. We appreciate the reviewer raising this concern. We have included in the discussion manuscript that we have published data showing that comparison within treatment groups can be trusted with this method.

  1. The method involves using gelatin as an embedding medium, which facilitates the simultaneous processing of multiple tissues. However, concerns have been raised regarding the integrity of gelatin sections containing thinner or irregularly shaped tissues. How does this affect the quality of histological sections, especially when detailed cellular structures need to be observed and analyzed?

Rebuttal: We have now included in the discussion how to mitigate the limitation with thinner sections – I think this is short shortcoming of the protocol to some degree or at least when compared to paraffin sections. This is particularly important for intestine tissue; however, for brain or other tissues, we have not found that 30 um sections limited the quality of the staining or analysis. The images used in the manuscript were sectioned at 30 um and the quality is maintained quite well. Thicker sections are employed more frequently now in neuroscience to model 3D structures (e.g., astrocytes end feet process on a vessel – this would be very difficult with thinner sections). One can also Z-stack the thick sections and compress the image file to a single image for analysis – this is typically needed for sections >40 um.

  1. The authors should elucidate the distinctions between gelatin and other embedding mediums in the Discussion section. Besides, it is crucial to highlight the advantages of using gelatin, providing a clear justification for its selection over alternative materials.

 Rebuttal: We thank the reviewer for this comment. We have highlighted the shortcomings of other methods in the introductory and discussion sections. And also discuss why our method is reliable (now highlighted in yellow in our revised manuscript). We also highlight in our discussion why this method is better compared to conventional IHC and other gelatin methods.  

  1. There are several formatting errors present in this manuscript. For instance, the symbol "℃" is incorrectly written as "C" (lines 84, 93, 94, 117, 125, etc.). Additionally, certain terms are inconsistently abbreviated, for example, "paraformaldehyde" is not abbreviated after its initial mention in line 84, while "RT" is abbreviated for the first time only in line 156, despite multiple prior occurrences. Please correct similar errors throughout the paper.

Rebuttal: We have addressed all occurrences/errors in the revised manuscript. We thank the reviewer for bringing this to our attention.

Reviewer 2 Report

Comments and Suggestions for Authors

The present manuscript describes a procedure for multiple embedding of mouse brains (or other histological specimens into gelatine. The block with embedded tissues can be frozen sectioned, and processed for various staining procedures. The procedure is not revolutionary, gelatin embedding techniques have been described in several papers. This protocol uses a highly concentrated solution of high strength gelatine, long fixation of blocks and a technical trick with cold block for arrangement of several species.  The procedure is clearly presented, and should be interesting for the readers of the journal. However, it should be noted that gelatin embedding has also shortcomings - there is relatively weak bond between the tissue and the matrix. As a consequence of that, thinner sections would disassemble. The same problem will be with smaller specimens. 

I appreciate detailed description of different staining protocols, as each of them has difficulties of its own.

I have three recommendations for improving this manuscript:

1, In Materials and Methods, section 2.4: Gelatin type A, size 300 – as far as I know this number is not size but strength of the gelatine (300 Bloom)

2, Albumin embedding alternatives should be mentioned in the Discussion 

3, What is the temperature of the cold block? Does it make the base and gelatin soaked specimens to freeze? Or is it just cooling that hardens the gelatine sufficiently to hold the specimen in place?

I recommend accepting this article after minor revisions.

Author Response

Reviewer 2:

The present manuscript describes a procedure for multiple embedding of mouse brains (or other histological specimens into gelatine. The block with embedded tissues can be frozen sectioned, and processed for various staining procedures. The procedure is not revolutionary, gelatin embedding techniques have been described in several papers. This protocol uses a highly concentrated solution of high strength gelatine, long fixation of blocks and a technical trick with cold block for arrangement of several species.  The procedure is clearly presented, and should be interesting for the readers of the journal. However, it should be noted that gelatin embedding has also shortcomings - there is relatively weak bond between the tissue and the matrix. As a consequence of that, thinner sections would disassemble. The same problem will be with smaller specimens.

I appreciate detailed description of different staining protocols, as each of them has difficulties of its own.

I have three recommendations for improving this manuscript:

1, In Materials and Methods, section 2.4: Gelatin type A, size 300 – as far as I know this number is not size but strength of the gelatine (300 Bloom)

Rebuttal: We thank the reviewer for this comment. We have addressed this in the updated manuscript.

2, Albumin embedding alternatives should be mentioned in the Discussion

Rebuttal: I found an article (PMID:22710286) that described this method; however, it is difficult to compare because the authors did not discuss its limitations – we briefly touched on this in the introductory section (now highlighted in yellow). Furthermore, we discussed other alternatives in the introduction (e.g., mega brain with OCT) and the brain array in the discussion section – we emphasized the advantage of our method over those methods. This is now highlighted in the manuscript for the reviewer. We have added the comments from the reviewer that the relatively weaker bonds between tissue and the matrix make it easy for thinner sections to disassemble – which is very true. We thank the reviewer for helping us make this clearer. Interestingly, this risk is greatest when less than 26-30 um. To prevent this, the technician can mount their sections after sectioning. This risk is greatest if the gelatin sections are stored > 2 months. Thicker sections do not typically have this issue in our experience. This is now all emphasized in the revised document and clearer.

3, What is the temperature of the cold block? Does it make the base and gelatin soaked specimens to freeze? Or is it just cooling that hardens the gelatine sufficiently to hold the specimen in place?

Rebuttal: The cooling block's temperature is 4C. It is just for cooling to harden the gelatin. We have clarified this in the text.